# Loss of CclA, required for histone 3 lysine 4 methylation, decreases growth but increases secondary metabolite production in *Aspergillus fumigatus*

Jonathan M. Palmer[1,4], Jin Woo Bok[1,4], Seul Lee[1,5], Taylor R.T. Dagenais[1,6], David R. Andes[1], Dimitrios P. Kontoyiannis[3] and Nancy P. Keller[1,2]

[1] Department of Medical Microbiology and Immunology, University of Wisconsin-Madison, Madison, WI, USA
[2] Department of Bacteriology, University of Wisconsin-Madison, Madison, WI, USA
[3] Department of Infectious Disease, University of Texas MD Anderson Cancer Center, Houston, TX, USA
[4] These authors contributed equally to this work.
[5] Current address: Chungbuk, Korea
[6] Current address: Madison Area Technical College, Madison, USA

Corresponding author
Nancy P. Keller, npkeller@wisc.edu

## ABSTRACT

Secondary metabolite (SM) production in filamentous fungi is mechanistically associated with chromatin remodeling of specific SM clusters. One locus recently shown to be involved in SM suppression in *Aspergillus nidulans* was CclA, a member of the histone 3 lysine 4 methylating COMPASS complex. Here we examine loss of CclA and a putative H3K4 demethylase, HdmA, in the human pathogen *Aspergillus fumigatus*. Although deletion of *hdmA* showed no phenotype under the conditions tested, the *cclA* deletant was deficient in tri- and di-methylation of H3K4 and yielded a slowly growing strain that was rich in the production of several SMs, including gliotoxin. Similar to deletion of other chromatin modifying enzymes, Δ*cclA* was sensitive to 6-azauracil indicating a defect in transcriptional elongation. Despite the poor growth, the Δ*cclA* mutant had wild-type pathogenicity in a murine model and the *Toll*-deficient *Drosophila* model of invasive aspergillosis. These data indicate that tri- and di-methylation of H3K4 is involved in the regulation of several secondary metabolites in *A. fumigatus*, however does not contribute to pathogenicity under the conditions tested.

## INTRODUCTION

*Aspergillus fumigatus* is a saprophytic filamentous fungus commonly found in diverse environments throughout the world. Whereas this fungus appears to have always been associated with aspergillomas since such structures were recognized by medical science in the 1800s, the incidence of invasive aspergillosis (IA) is a more recent phenomenon directly correlated to the increase in immune-compromised patients originally treated for other

diseases. Why *A. fumigatus* predominates as the causative agent of IA instead of the 200 or so other Aspergilli is a question of considerable interest as the answers should help in devising strategies to control this often fatal disease.

*A. fumigatus* possesses several attributes that are thought to contribute to its predominance in IA and other *Aspergillus*-associated diseases; this topic has been the subject of several recent reviews (*Dagenais & Keller, 2009*; *Abad et al., 2010*; *Mccormick, Loeffler & Ebel, 2010*). The reviews indicate that this species may not possess unique classical virulence factors, but rather orchestrates the expression of conserved fungal characteristics in such a way as to evade or overcome host defenses more so than other Aspergilli. There are, however, at least two attributes that have been associated with *A. fumigatus* virulence that vary considerably among *Aspergillus* species: its remarkable thermo-tolerance and its unique repertoire of toxic secondary metabolites (SM) (*Dagenais & Keller, 2009*; *Abad et al., 2010*).

Many studies dating back almost 30 years have implicated several toxins as important in host – *Aspergillus* interactions (*Mullbacher, Waring & Eichner, 1985*; *Eichner et al., 1986*; *Sutton et al., 1994*; *Amitani et al., 1995*; *Khoufache et al., 2007*; *Fallon, Reeves & Kavanagh, 2011*). In recent years, individual genes required for SM production have been removed from the *A. fumigatus* genome and the resultant deletants tested for virulence in various animal models (*D'enfert et al., 1996*; *Reeves et al., 2006*; *Ben-Ami et al., 2009*). Through these studies it was discovered that a global regulator of SM production, LaeA, was shown to be important in IA development (*Bok et al., 2005*; *Sugui et al., 2007a*). Furthermore, the LaeA regulated SM gliotoxin contributes to *A. fumigatus* virulence in several animal models of IA (*Reeves et al., 2004*; *Stanzani et al., 2005*; *Cramer et al., 2006*; *Coméra et al., 2007*; *Sugui et al., 2007b*; *Ben-Ami et al., 2009*; *Lionakis & Kontoyiannis, 2010*). Since then, global regulation of SM gene clusters has been found to be regulated in part through global changes in histone modifications, particularly acetylation and methylation of histone 3 tail residues (*Bok et al., 2005*) and reviewed in *Strauss & Reyes-Domínguez (2011)*. These modifications have been linked to LaeA activity (*Reyes-Dominguez et al., 2010*). A recent report in *A. nidulans* showed that CclA, a member of the conserved eukaryotic COMPASS complex that methylates histone 3 lysine 4 (H3K4), was also critical in the regulation of several SMs (*Bok et al., 2009*). H3K4 can have up to five different modifications: acetylated, unmodified, mono-methylated, di-methylated, and tri-methylated. Histone tail modifications were once thought to be permanent, however more recent data has indicated that these modifications are dynamic, *i.e.* they are reversible. For example, while the COMPASS complex methylates H3K4 there is a reciprocal enzyme that demethylates H3K4. In mammalian systems, a flavin containing amine oxidase named LSD1 (lysine specific demethylase 1) is responsible for demethylation of H3K4 (*Shi et al., 2004*).

Based on our previous research in the model fungus *A. nidulans*, we identified the *A. fumigatus cclA* and LSD1 homologs from the genome sequence. To investigate a possible role of H3K4 methylation in SM cluster regulation and pathogenicity in *A. fumigatus*, we deleted both *cclA* and the putative LSD1 homolog, which we named *hdmA* (histone demethylase A). Whereas the *hdmA* deletant was indistinguishable from

**Table 1** Strains used in this study.

| Strain | Genotype | Source |
|--------|----------|--------|
| AF293 | Wild type | (*Xue et al., 2004*) |
| AF293.1 | *pyrG1* | (*Xue et al., 2004*) |
| AF293.6 | *pyrG1, argB1* | (*Xue et al., 2004*) |
| TJW84.1 | *pyrG1, ΔcclA::A. parasiticus pyrG* | This study |
| TJW88.21 | *pyrG1, ΔcclA::A. parsasiticus pyrG, cclA::hygromycin* | This study |
| TJMP2.36 | *pyrG1, argB1, ΔhdmA::A. parasiticus pyrG* | This study |
| TJMP3.52 | *pyrG1, argB1, ΔhdmA::A. parasiiticus pyrG, A. fumigatus argB* | This study |
| TJMP29.1 | *pyrG1, argB1, ΔhdmA::A. parasiticus pyrG, hdmA:A. fumigatus argB* | This study |

wild type, the *cclA* deletant, similar to the phenotype of the *A. nidulans ΔcclA* mutant, was deficient in tri- and di-methylation of H3K4 and exhibited slow growth but increased activation and expression of several *A. fumigatus* SMs, including gliotoxin. The mutant showed wild type pathogenicity in both a neutropenic murine model and *Toll*-deficient *Drosophila* model of invasive aspergillosis. These data suggest that tri- and di-methylation of H3K4 is not required for pathogenicity under the conditions tested and raises the possibility that pathogenicity of a slow growing mutant strain may be compensated by enhanced gliotoxin synthesis.

## MATERIALS AND METHODS

### Strains, growth conditions, and DNA manipulations

All *A. fumigatus* strains used in this study are derivatives of the clinical isolate AF293 (*Xue et al., 2004*) and are listed in Table 1. Strains were maintained on glucose minimal media (GMM) (*Shimizu & Keller, 2001*) and when appropriate the media was supplemented with 5 mM uracil and 5 mM uridine for *pyrG1* strains, 5 mM arginine for *argB1* strains, and hygromycin was used at a final concentration of 100 ug/ml. Radial growth experiments and 6-azauracil sensitivity assays were carried out essentially as in *Palmer et al. (2008)*. All DNA manipulations were conducted according to standard protocols (*Sambrook & Russell, 2001*) and fungal transformation was done as previously described (*Miller, Miller & Timberlake, 1985*) with a minor modification of protoplasts being plated in 0.75% molten top agar. Primers used in this study are listed in Table 2.

### Construction of mutant *A. fumigatus* strains

An *A. fumigatus cclA* disruption cassette was constructed using double-joint PCR consisting of the following: 1-kb DNA fragment upstream of the *cclA* start codon (primers Fbre5F and Fbre5R-*Eco*RI), a 3-kb *Eco*RI-*Hind*III DNA fragment of *A. parasisitus pyrG* via pJW24 (*Calvo et al., 2004*), and a 1-kb DNA fragment downstream of the *cclA* stop codon (primers Fbre3F and Fbre3R-*Hind*III). The fusion PCR product was then used to transform AF293.1 to prototrophy creating TJW84.1. Homologous single-gene replacement of *cclA* was confirmed by Southern analysis (Figs. 1A and 1B). In order to complement the *ΔcclA* mutant, a complementation plasmid (pJW109.5) was constructed.

**Table 2** Primers used in this study.

| Name | Sequence (5′ to 3′): Restriction sites underlined | Purpose |
|---|---|---|
| Fbre5F | ATATCAGTTGTTGCTCCTAGGGC | 5′ Flank Δ*cclA* |
| Fbre5R-*Eco*RI | CATGTGGAATTCAATAACGGTTCACGAGTAAATTG | 5′ Flank Δ*cclA* |
| Fbre3F-*Hind*III | TGGGGTAAGCTTGTTCCGAGCCATATCTGTC | 3′ Flank Δ*cclA* |
| Fbre3R | ACAACAAAACTAGCTCTCTCGGC | 3′ Flank Δ*cclA* |
| FbreCOMF-NotI | TTGCTCCAGCGGCCGCATCCTCAAACCTCGCCAAGTATG | *cclA* complementation |
| FbreCOMR-SpeI | GGAAATACTAGTGAAGCTGAAAGTGACGTCTAGAA | *cclA* complementation |
| JP1 | TATAGGGACGAATTCACAGACAATG | 5′ Flank Δ*hdmA* |
| JP2 | CGCTGAATTCGGGTTTGATGGACATTGGAC | 5′ Flank Δ*hdmA* |
| JP3 | CCGGTTCTAGAGGTAATGCTTAGACTCCCGTA | 3′ Flank Δ*hdmA* |
| JP4 | GTCATGCGGCCGCCACTGCCCTCGTTAAGG | 3′ Flank Δ*hdmA* |
| JP lsdA comp For | GCTTGGATCCGACGTAGCAGGTGAAC | *hdmA* complementation |
| JP lsdA comp Rev | CATTGGATCCTCCTCGCCTTTCTCC | *hdmA* complementation |
| JP Afumi argB For | GAACGCGGTCTGCATCCAAG | *Af-argB* cloning |
| JP Afumi argB Rev | GAAGGAGAGACCCATACATCC | *Af-argB* cloning |
| FumlaeAF | ATGTTTCTCAACGGGCAGGGC | *laeA* probe |
| FumlaeAR | ATTGGCGAGAGGTTTTCGAGCC | *laeA* probe |
| FbreIF | CTTTCCACACATCAAGTACCGGC | *cclA* probe |
| FbreIR | ATTGAAGCGTTCGCCAATGCC | *cclA* probe |
| JP LSD1 | TCTCACGCAACTACATACGTCAAC | *hdmA* probe |
| JP LSD2 | TTTTTCCTCTTCGCTGGCTTGCC | *hdmA* probe |
| GZINTF | AAGGGCCGGTAGTCTACCTCTTC | *gliZ* probe |
| GZINTF | CGATCTGGTAGCTGCCCAGCTGGAAG | *gliZ* probe |

The plasmid contained a 3.8-kb wild-type *cclA* gene including a 1-kb native promoter and a 1-kb native termination cassette, which was amplified by primers FbreCOMF-NotI and FbreCOMR-SpeI. The PCR product was cloned into a *NotI-SpeI* site of pUCH2-8 (*Alexander, Hohn & Mccormick, 1998*), which contains the selectable marker hygromycin B phosphotransferase. TJW84.1 was transformed with pJW109.5 to hygromycin resistance and Southern blot confirmed single copy integration for TJW88.21 (Fig. 1B).

An *hdmA* disruption cassette was constructed by PCR amplifying a 1.1-kb DNA fragment upstream of the *hdmA* ORF using the primer pair JP1 and JP2 containing *EcoRI* restriction sites. The subsequent 1.1-kb *EcoRI-EcoRI* fragment was cloned into pJW24 to create pJMP1. A downstream 1.0-kb fragment was amplified using the primer pair JP3 and JP4 containing *XbaI* and *NotI* restriction sites respectively. The 1.0-kb *XbaI–NotI* fragment was cloned into pJMP1 to create the *hdmA* disruption cassette named pJMP2. The *hdmA* ORF was disrupted by transformation of linearized pJMP2 into AF293.6 creating the Δ*hdmA argB1* auxotrophic strain (TJMP2.36), which was confirmed by Southern blot (Figs. 1C and 1D). In order to generate a prototrophic deletion strain, an *argB* complementation plasmid was constructed by amplifying a 2.6-kb fragment corresponding to *A. fumigatus argB* using the primer pair 'JP Afumi argB For' and 'JP Afumi argB Rev', which was subsequently cloned into pCR2.1 TOPO (Invitrogen) to

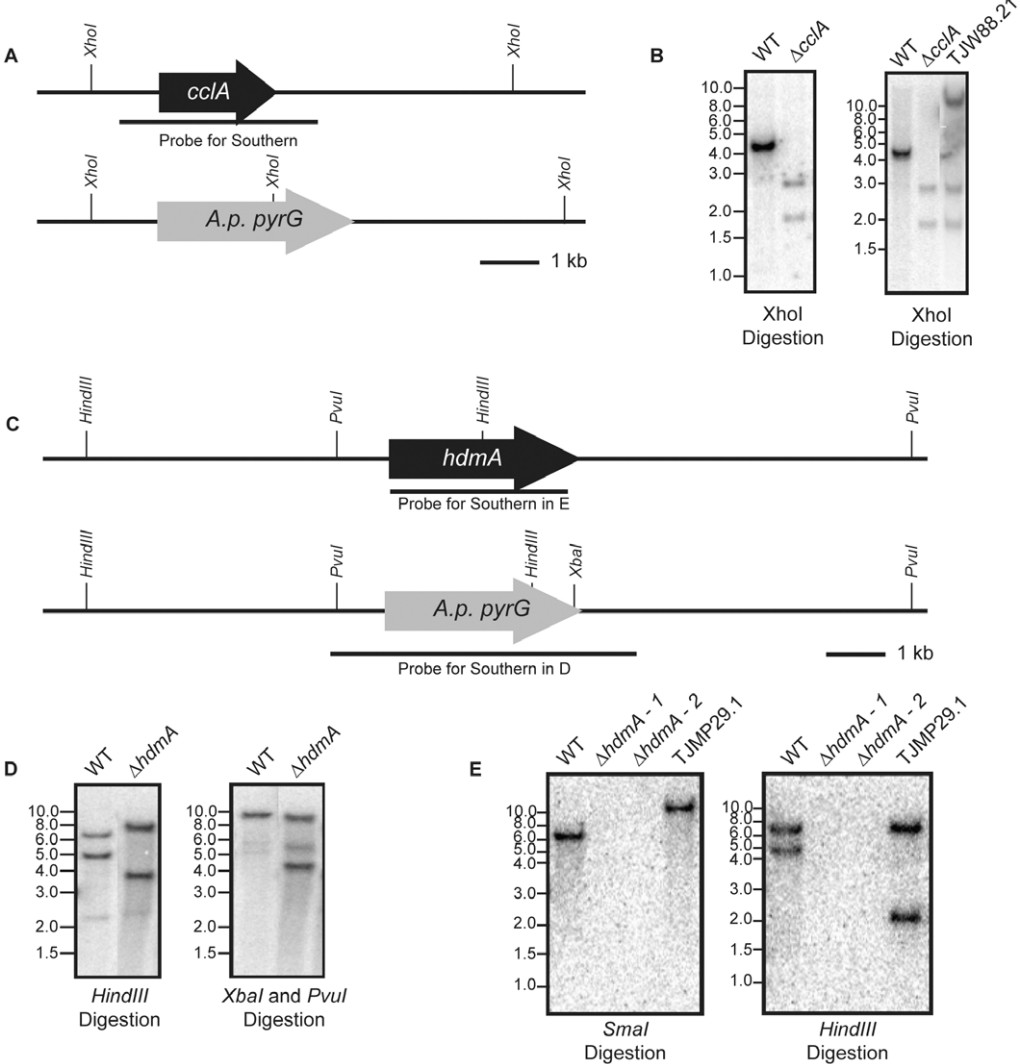

**Figure 1 Construction of deletion and subsequent complementation strains of *cclA* and *hdmA* in *A. fumigatus.*** (A) A schematic drawn to scale illustrates the replacement of the *cclA* ORF with the *A. parasiticus pyrG* gene and the location of *XhoI* restriction enzyme sites. (B) A radiolabeled probe consisting of the *cclA* ORF and ∼1 kb on either side was used to probe a Southern blot of *XhoI* digested genomic DNA. The expected banding pattern was observed. Confirmation of a complemented *cclA* strain (TJW88.21) was achieved via Southern blot using the described conditions above. An additional band is shown that is greater than 10 kb that is attributed to the random integration of the *cclA* complementing plasmid. (C) Schematic drawn to scale of the restriction enzyme cut sites in the wild type and Δ*hdmA* backgrounds. (D) Southern blot using a *Hind*III digestion and a *Xba*I–*Pvu*I double digestion using a radiolabeled probe consisting of the gene deletion construct shows the expected size differences between wild type and the Δ*hdmA* strains. (E) Complementation of the Δ*hdmA* mutant was confirmed using a *Sma*I and subsequent *Hind*III digestion to determine single integration of the complementing plasmid (TJMP29.1).

create pJMP4. This plasmid was transformed into TJMP2.36 to generate a prototrophic Δ*hdmA* strain (TJMP3.52). For complementation of Δ*hdmA*, a 4.6-kb fragment corresponding to ∼0.9-kb up stream and ∼0.5-kb downstream of the *hdmA* ORF was
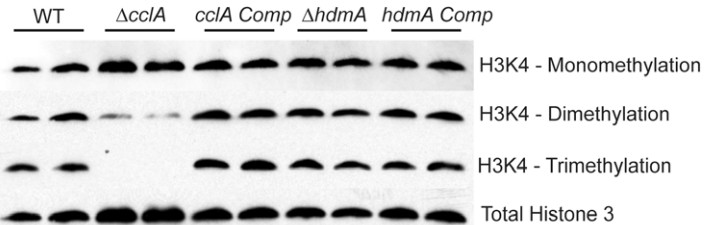

**Figure 2 Deletion of *cclA* results in a reduction in di-methylation and loss of tri-methylation of lysine 4 of histone 3 (H3K4).** Western blot of nuclear protein extracts clearly indicate that *cclA* is required for proper methylation of H3K4. While H3K4 mono-methylation is unaffected by loss of *cclA*, di-methylation is reduced and tri-methylation was absent. These data are consistent with *S. cerevisiae BRE2* null mutants. No difference in global methylation patterns of H3K4 was detected in the Δ*hdmA* strain compared to wild type.

amplified with the primer pair 'JP lsdA comp For' and 'JP lsdA comp Rev' each containing a *BamHI* site. The subsequent *BamHI-BamHI* fragment was cloned into pJMP4 to create pJMP13. TJMP2.36 was transformed with pJMP13 to create a complemented control strain TJMP29.1 (Δ*hdmA* + *hdmA*) and single integration was confirmed by Southern blot (Fig. 1E).

## Physiology experiments

Radial growth measurements and sensitivity to 6-azauracil (6AU) was conducted as described previously (*Palmer et al., 2008*). Four replicates were used for each assay and statistical significance was calculated with ANOVA analysis using Prism 5 software (Graph Pad).

## Alterations in H3K4 methylation

Nuclear extracts were isolated as previously described (*Palmer et al., 2008*). Approximately 50 µg of nuclear protein extract was electrophoresed on a 10% Tricine-SDS-PAGE gel (*Schägger, 2006*) and subsequently electroblotted to nitrocellulose membranes. Detection of H3K4 modifications was conducted using the following primary antibodies and dilutions: 1:1,000 anti- histone 3 (Upstate, #07-690), 1:1,000 anti-H3K4 mono-methylation (Upstate, #07-436), 1:2,000 anti-H3K4 di-methylation (Upstate, #07-030), 1:2,000 anti-H3K4 tri-methylation (Upstate, #07-473). Chemiluminescent detection was employed using SuperSignal West Pico Chemiluminescent Substrate (Thermo Scientific) and a secondary goat anti-rabbit-horseradish peroxidase conjugate antibody (Pierce, #31460) diluted 1:15,0000 (Fig. 2).

## Northern analysis and secondary metabolite extraction

Fifty milliliter cultures of liquid GMM + 0.1% yeast extract were inoculated with $1 \times 10^7$ spores per ml and incubated at 250 rpm 29 °C for 24 h, followed by a reduction in temperature to 25 °C for an additional 48 h. Mycelia were harvested, lyophilized overnight, and total RNA was extracted using Isol-RNA Lysis Reagent (5 Prime) according to manufacturers recommendations. Subsequent northern analysis was done using radiolabeled probes for the corresponding transcript (Fig. 4C) (primers are listed

in Table 2). Secondary metabolites were extracted from 20 ml of culture filtrate with an equal volume of chloroform. The air-dried chloroform layer was resuspended in 75 µl chloroform and 5 µl was separated on a thin layer chromatography plate (Whatman, #4410 222) using chloroform: acetone (7:3) as a solvent. The TLC plate was then dried and imaged under 254 nm and 366 nm ultraviolet light (Fig. 4A). Organic extracts from solid minimal medium cultures was accomplished by taking equal sized cores from the center of point-inoculated cultures, homogenized in 3 ml of water, and extracted with an equal volume of chloroform. Quantification of TLC spots was achieved through densitometry analysis using Image J and normalized to wild type levels.

## Pathogenicity assays

The neutropenic murine model of aspergillosis was conducted as described previously (*Bok et al., 2005*) with approval of the William S. Middleton VA and the University of Wisconsin Animal Care Committees in accordance with the MV2344 animal use protocol. *Toll*-deficient flies were generated by crossing flies carrying a thermosensitive allele of *Toll* (Tl r632) with flies carrying a null allele of *Toll* (Tl I-RXA) (*Lionakis et al., 2005*). Two- to four day old adult female *Toll*-deficient flies were used in all of the experiments. Twenty flies were infected with each *A. fumigatus* strain used in this study. All of the experiments were performed in triplicate. To minimize **circadian** rhythm variability, all experiments were performed at 9 AM. *A. fumigatus* isolates were grown on yeast extract agar glucose (YAG) at 37 °C. Conidia were collected in sterile 0.9% saline from 2 days old cultures. The conidial concentration suspension was determined by using a hemacytometer and adjusted to $1 \times 10^7$ per ml. The dorsal side of the thorax of twenty $CO_2$ anesthetized flies was punctured with a thin (10 µm) sterile needle that had been dipped in a concentrated solution of *A. fumigatus* conidia ($10^7$ cells/ml). In our previous work, this method was shown to deliver a reproducible inoculum of 700 to 800 conidia (*Lionakis et al., 2005*). As a negative control group, *Toll*-deficient flies were punctured with a 10 µm sterile needle and monitored daily for survival. Flies that died within 3 h of the injection were considered to have died as a result of the puncture procedure and were not included in the survival rate analysis. The flies were housed in a 29 °C incubator to maximize expression of the Tl r632 phenotype (*Lemaitre et al., 1996*). The *Toll*-deficient flies were transferred into fresh vials every 3 days. Fly survival was assessed daily over 8 days.

## RESULTS

### Deletion of *cclA* but not *hdmA* affects global histone 3 lysine 4 methylation patterns

*Aspergillus fumigatus cclA* was identified through BLASTp analysis of the *A. fumigatus* genome with the *A. nidulans* CclA amino acid sequence (*Bok et al., 2009*), which yielded one homolog Afu3g04120. The human LSD1 protein sequence (Genbank, #O60341) was used to BLASTp the *A. fumigatus* genome sequence, which provided one potential ortholog, Afu4g13000. Afu4g13000 is 29% identical to human LSD1 and contains the

conserved SWIRM domain, a flavin amine oxidase domain, as well as an HMG Box domain (*Shi et al., 2004*) and thus was named *hdmA* for histone demethylase A.

The *cclA* and *hdmA* alleles were independently disrupted in the *A. fumigatus* AF293 genetic background. Southern analysis of transformants for each gene confirmed simple gene replacements (Fig. 1). One representative deletant, TJW84.1 for Δ*cclA* and TJMP3.52 for Δ*hdmA*, was chosen for each gene replacement for the studies presented below. Both deletants were complemented with their respective genes to generate a single copy *cclA* + Δ*cclA* strain and a single copy *hdmA* + Δ*hdmA* strain as shown in Table 1.

Our first experiment was to assess if loss of either gene affected histone 3 lysine 4 methylation (H3K4). The Δ*cclA* strain showed a clear loss of whole genome H3K4 tri-methylation and a considerable decrease in di-methylation (Fig. 2); this tallied with earlier studies in our lab assessing H3K4 methylation in an *A. nidulans* Δ*cclA* mutant (*Bok et al., 2009*). These data are also consistent with *BRE2* mutants in *Saccharomyces cerevisiae*, where loss of *BRE2* results in strains with diminished di-methylation and no tri-methylation of H3K4 (*Schneider et al., 2005*). The complemented strain was restored in H3K4 methylation to wild type levels. However, loss of *hdmA* did not exhibit a detectable impact on whole genome H3K4 methylation (Fig. 2).

## CclA mutants are crippled in growth and sensitive to 6-Azauracil

As reported for the *A. nidulans* Δ*cclA* mutant (*Giles et al., 2011*), the *A. fumigatus* Δ*cclA* strain exhibited poor growth as exhibited by decreased radial growth on several different media, including rich media (YPD and Champs) as well as our standard minimal medium (GMM) (Fig. 3A). Additionally, the Δ*cclA* mutant had a measurable growth defect at all temperatures tested (Fig. 3B) and decreased mass when grown in liquid shaking culture (data not shown). This defect was rescued in the complemented strains. Neither the *hdmA* deletant nor its complement exhibited any growth phenotype.

The phenotype of the Δ*cclA* strain shared some similarities to that of a previously described *A. fumigatus* chromatin mutant, Δ*clrD* (*Palmer et al., 2008*), impaired in methylation of H3K9. As histone methyltransferase mutants are often found to be sensitive to chemicals that target various cellular processes, the Δ*clrD* mutant had been assessed for sensitivity to various chemical agents and found to be sensitive to 6-azauracil (6AU); an inhibitor of guanine nucleotide synthesis and indicator of transcriptional defects (*Riles et al., 2004*; *Zhang et al., 2005*; *Palmer et al., 2008*). Following that protocol, the Δ*cclA* and Δ*hdmA* strains were similarly screened and the former but not the latter found to be also sensitive to 6AU (Fig. 3C). Resistance to 6AU was rescued in the complement strain. Moreover, we did not find differences in sensitivity of Δ*hdmA* or Δ*cclA* strains compared to wild type when exposed to several chemical agents that included hydroxyurea (DNA synthesis inhibitor), thiabendazole (microtubule destabilizer), 1.2 M Sorbitol (osmotic stress), 0.6 M KCl (osmotic stress), Congo red (cell wall inhibitor), or Calcofluor white (cell wall inhibitor) (data not shown).

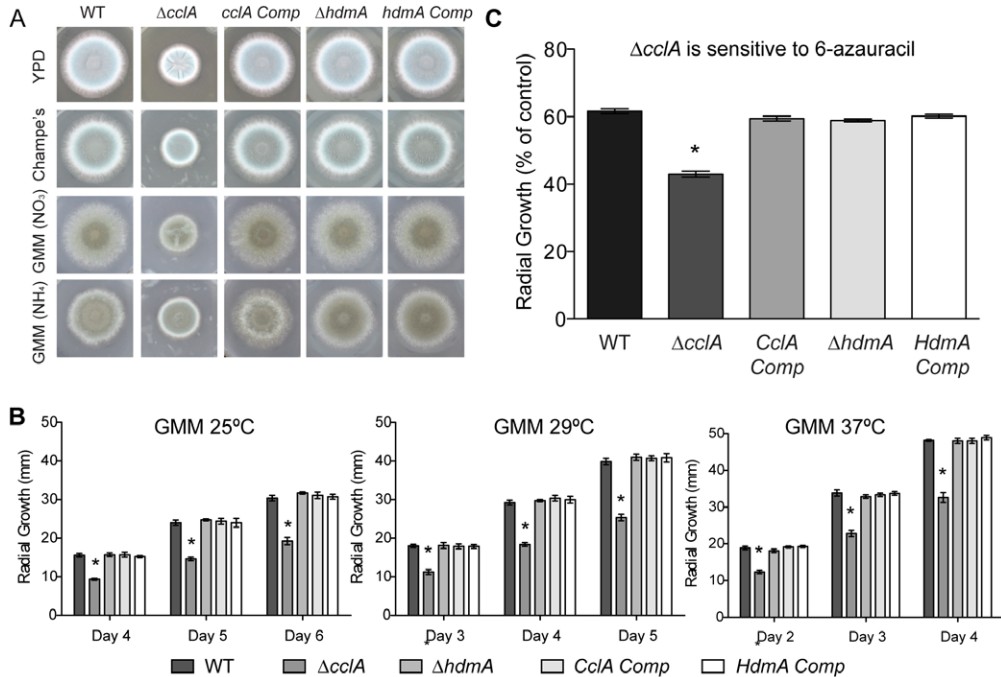

**Figure 3 Null mutants of *cclA* result in reduced growth and increased sensitivity to 6-azauracil.** (A) The Δ*cclA* strain is defective in radial growth on rich media (YPD, Champs) as well as minimal media (GMM) using both nitrate and ammonium as a nitrogen source. The complemented control strain restores wild-type levels of growth. (B) Quantification of radial growth on solid media at three different temperatures (25 °C, 29 °C, and 37 °C) illustrates a significant growth reduction in the Δ*cclA* strain in all conditions tested. No phenotype was observed for a mutant lacking *hdmA*. (C) Radial growth assays conducted on GMM amended with 100 ug/ml of 6-azauracil indicated that null mutants of *cclA* are more sensitive to 6-azauracil than wild type. The *cclA* complemented control strain restores sensitivity to 6-azauracil to wild type levels. Asterisk indicates $p < 0.001$ using an ANOVA test for statistical significance with Prism 5 software.

## CclA loss results in increased secondary metabolism

Mutations in chromatin remodeling genes have frequently been associated with alterations in secondary metabolite production, both in *A. nidulans* (*Shwab et al., 2007*; *Bok et al., 2009*) and *A. fumigatus* (*Lee et al., 2009*). In particular the *A. nidulans* Δ*cclA* strain was found to produce novel metabolites with antimicrobial properties (*Bok et al., 2009*; *Giles et al., 2011*). An examination of extracts from three different conditions of the two mutants showed that the *A. fumigatus* Δ*cclA* strain but not the Δ*hdmA* strain was increased in the production of several metabolites (Fig. 4). In particular, this strain produced over 4 times as much gliotoxin compared to wild type when grown in liquid shaking culture (Figs. 4A and 4B). The high level of gliotoxin was associated with a large increase in transcription of *gliZ*, the C6 transcription factor required for *gli* expression (Fig. 4C). Under these growth conditions, loss of *cclA* also resulted in several fold increases of numerous other metabolites (Figs. 4A and 4B). The Δ*cclA* mutant was also analyzed under alternative growth conditions and produced similar amounts of metabolites on solid

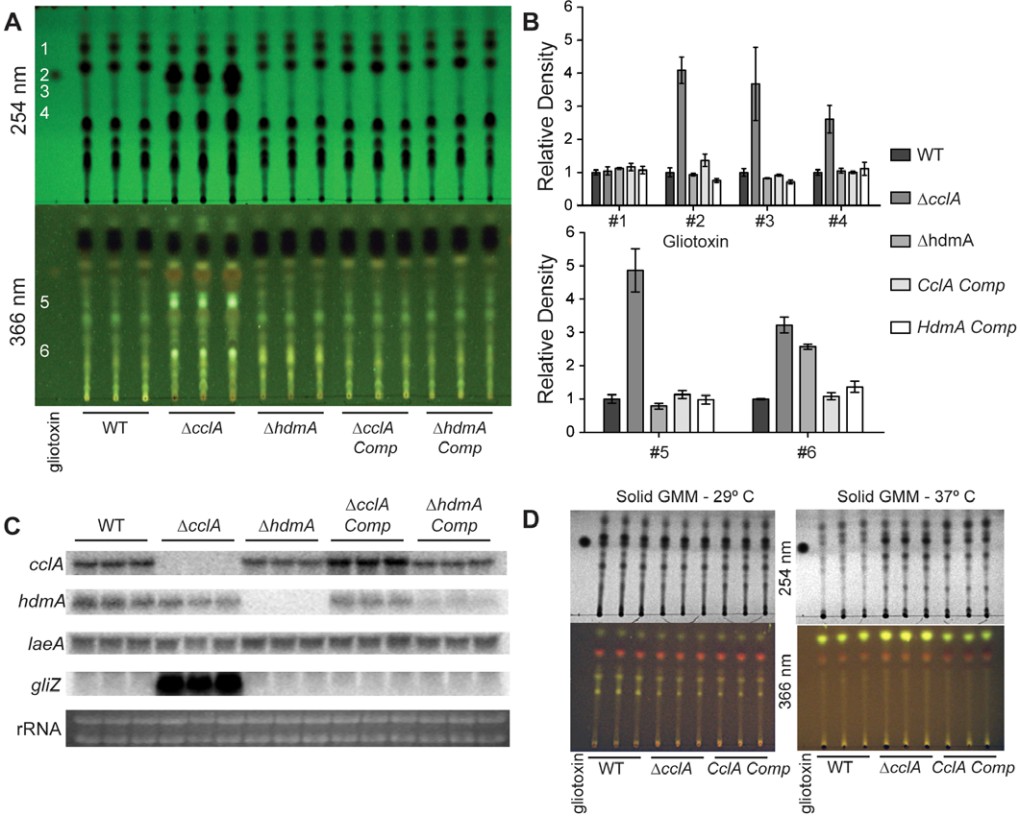

**Figure 4 CclA, but not HdmA, controls production of several secondary metabolites, including gliotoxin.** (A) Strains were grown in liquid minimal media shaking culture and analyzed for production of secondary metabolites by thin layer chromatography (TLC) using gliotoxin as a standard. UV infused TLC plates were photographed under 254 nm and 366 nm light. (B) Quantification of TLC spots was achieved by densitometry with Image J software and relative density was normalized to wild type extracts, which reveals that null mutants of *cclA* produced more than 4 times more gliotoxin as well as several fold more production of several unidentified metabolites. (C) Northern analysis from the same growth conditions confirms the strains as well as identifies a large increase in *gliZ* transcript in the Δ*cclA* mutant background compared to wild type. There are no differences in transcription of the regulator of secondary metabolism *laeA*. (D) Organic extracts were prepared from cultures grow on solid media and subsequently analyzed via TLC analysis using chloroform:acetone (7:3) as a solvent. Growth at 29 °C results in no detectable difference between Δ*cclA* and wild type, however at 37 °C a few metabolites are slightly increased in Δ*cclA* compared to wild type.

medium at 29 °C compared to wild type (Fig. 4D), however Δ*cclA* mutants produce more of a few unidentified metabolites when grown 37 °C on solid medium (Fig. 4D).

## Virulence of Δ*cclA* is unaltered in a mammalian and *Drosophila* model

Although poorly growing *A. fumigatus* mutants have often been found to exhibit a decrease in virulence (*D'enfert et al., 1996*; *Brown et al., 2000*), the increased toxin production in the Δ*cclA* strain coupled with the poor growth presented a unique phenotype that disallowed an obvious prediction on its pathogenicity attributes. Therefore, to assess *cclA* or *hdmA* loss on virulence, we examined virulence in two animal models. Assessment of the mutants in

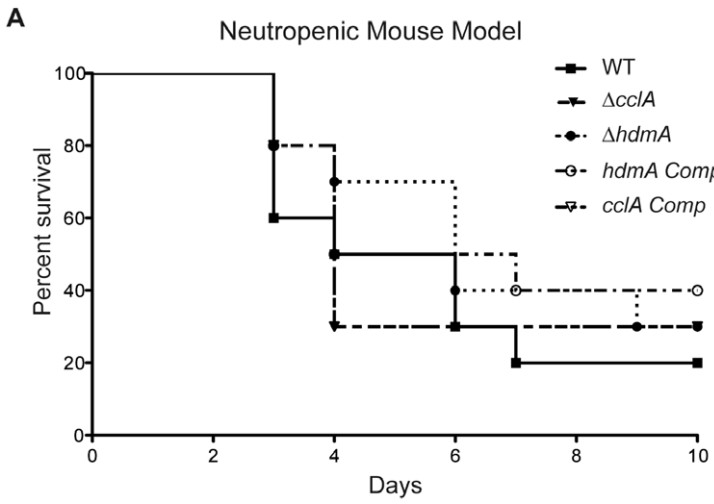

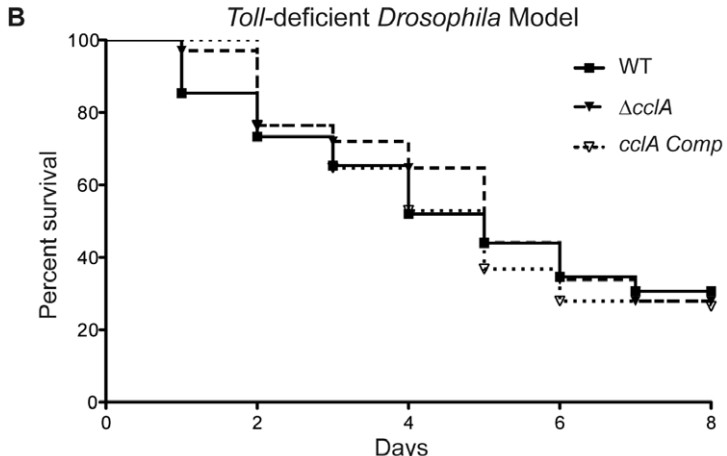

**Figure 5 Pathogenicity of ΔcclA and ΔhdmA strains are wild type in murine model of IA and ΔcclA strains are wild type in *Toll*-deficient *Drosophila* model.** (A) Pathogenicity of WT, ΔcclA, cclA complement, ΔhdmA, and hdmA complement strains was assessed using the neutropenic murine model of aspergillosis. No differences were measured in any of the strains tested in comparison to wild type (AF293). (B) An alternative pathogenicity assay using *Toll*-deficient fruit flies illustrates that the ΔcclA mutant is no different than wild type. The Mantel–Cox statistical test was used to assess differences in survival curves.

a neutropenic murine model showed no significant difference between the wild type and mutant strains (Fig. 5A). This was also true for the *Toll*-deficient *Drosophila* model (where only ΔcclA was assessed, Fig. 5B).

## DISCUSSION

Genes involved in secondary metabolism in fungi are arranged in a clustered format that is easily impacted by chromatin level alterations (*Palmer & Keller, 2010*). This characteristic has been illustrated primarily by two ways: either through manipulation (deletion or overexpression) of chromatin modifying genes or SM induction via

chemical epigenetics (*Bok et al., 2009*; *Cichewicz, 2010*). LaeA, a global regulator of SM (*Reyes-Dominguez et al., 2010*), is a conserved virulence factor in filamentous fungi including *A. fumigatus, A. flavus* and *Gibberella zeae* (*Bok et al., 2005*; *Sugui et al., 2007a*; *Amaike & Keller, 2009*). Although the mechanism of LaeA action is not known, the results of its activity, such as decrease in heterochromatin marks (e.g. H3K9 tri-methylation, (*Reyes-Dominguez et al., 2010*)) correlated with increased SM cluster expression, are thought to contribute to its role as a virulence factor in IA. Extrapolation of LaeA impact on chromatin led to the hypothesis that other chromatin remodeling genes, such as histone methyltransferases, acetylases and their cognate demethylases and deacetylases would also impact SM production and, possibly, virulence attributes in pathogenic fungi such as *A. fumigatus*.

Here our efforts focused on two conserved genes that in other systems are involved in methylation and demethylation of histone 3 on lysine 4. The methylation state of H3K4 is associated with both gene activation and repression in eukaryotes (*Krogan et al., 2002*; *Mueller, Canze & Bryk, 2006*). The conserved eukaryotic COMPASS complex is required for methylation of H3K4 with one of its key members a SPRY domain protein termed Bre2p in *Saccharomyces cerevisiae* (*Krogan et al., 2002*), Ash2p in *Schizosaccharomyces pombe* (*Roguev et al., 2001*) and CclA in *A. nidulans* (*Bok et al., 2009*). Demethylation of H3K4 is achieved through activity of an amine oxidase called LSD1 in higher eukaryotes (*Shi et al., 2004*; *Shi et al., 2005*). *Aspergillus* species contain one putative LSD1 homolog, which we termed *hdmA* in this work due to the existence of a previously named *lsdA* gene – containing no similarity to *hdmA* – involved in late sexual development in *A. nidulans* (*Lee et al., 2001*).

Our previous results in *A. nidulans* demonstrated a critical role for CclA in both SM production as well as normal growth (*Bok et al., 2009*; *Giles et al., 2011*). This phenotype was replicated in the *A. fumigatus* Δ*cclA* mutant. On the other hand, the *hdmA* deletant in *A. nidulans* had a more subtle effect, and similar to what we describe here for the *A. fumigatus* Δ*hdmA* mutant, small impact on fungal morphology (data not shown). Notably, deletion of *cclA* in either species was accompanied by an easily detectable decrease in H3K4 methylation, however under the conditions used in this study, no difference in H3K4 methylation was observed in the Δ*hdmA* strain. Deletion or down regulation of this protein in other organisms can be detected by increased H3K4 methylation (*Shi et al., 2004*); possibly this increased H3K4 methylation is not observable in bulk histone assessment as measured in this study or, alternatively, HdmA is not the (major) H3K4 demethylase in *A. fumigatus*. An additional class of Jumonji C domain (JmjC) containing proteins has been shown to demethylate lysine residues in histone tails. While chromatin structure in the aspergilli is thought to be more similar to fission yeast and higher eukaryotes, in *S. cerevisiae* a JmjC domain containing protein, Jhd2p, is the major demethylase of H3K4 (*Liang et al., 2007*; *Seward et al., 2007*). Moreover there is a Jhd2 homolog (Afu5g03430) present in the *A. fumigatus* genome and thus perhaps Afu5g03430 is the major demethylase of H3K4 in the aspergilli.

Stress assays suggested that the Δ*cclA* strain was more sensitive to 6AU, a chemical used to identify strains impaired in transcriptional processes, than wild type or the complemented strain. Use of this chemical has identified several *S. cerevisiae* histone methyltransferase mutants defective in transcriptional elongation (*Exinger & Lacroute, 1992*; *Li, Moazed & Gygi, 2002*; *Zhang et al., 2005*), therefore it is perhaps not surprising that two *A. fumigatus* histone methyltransferase mutants – Δ*cclA* described in this study and Δ*clrD* previously described – are also sensitive to this chemical. Whereas the mechanism of Δ*cclA* sensitivity to 6AU is unknown, this sensitivity could reflect a defect in the fungal transcriptional machinery in this strain, which highlights the pleiotropic phenotypes of these mutants.

The increased SM output from the Δ*cclA* strain (Fig. 4) likely arises from aberrancies in transcriptional activity, in this case primarily enhancing transcription of a subset of SM clusters as previously demonstrated in the *A. nidulans* Δ*cclA* mutant (*Bok et al., 2009*). The increased expression of *gliZ* correlated well with the significant increase in gliotoxin synthesis in this strain. Although we have not characterized the other up regulated metabolites in this strain, we speculate that some of them may be a result of 'turning on' formerly silent SM clusters and we will be investigating this hypothesis in future studies.

In *A. fumigatus*, mutants that have been shown to display poor growth *in vitro* have also been shown to be less virulent in a model of IA (*D'enfert et al., 1996*; *Brown et al., 2000*). However, it was recently shown in *Candida albicans* that mutants that grow poorly *in vitro* do not always show a decrease in virulence (*Noble et al., 2010*). Similarly, disruption of ClrD in *A. fumigatus* results in a mutant that is defective in growth and asexual sporulation (*Palmer et al., 2008*) however did not result in a decrease in pathogenicity in the mouse model of IA (T. Dagenais, D. Andes & N. Keller, unpublished data). Thus, the pleiotropic phenotype of poor growth in the laboratory with increased secondary metabolite synthesis of the Δ*cclA* mutant presented an interesting and potentially opposing coupling of putative *A. fumigatus* virulence attributes, as enhanced gliotoxin production would be expected to increase virulence (*Bok et al., 2006*; *Cramer et al., 2006*; *Sugui et al., 2007b*). The results from our studies suggest that loss of the CclA gene had no significant effect on pathogenicity in either a neutropenic murine or *Drosophila* model of IA. Although we did not test growth rate in the host organisms, our data could suggest that pathogenicity can be restored in poor growing strains by increased secondary metabolite (gliotoxin) production, or alternatively, could indicate that genetic factors regulating *in vitro* growth may differ from those controlling *in vivo* growth. Moreover, this study supports the view that a composition of many *A. fumigatus* characteristics contributes to the pathogenicity of this species.

### Funding

This research was funded by National Institutes of Health (NIH), National Research Service Award AI55397 to JMP and by NIH 1 R01 Al065728-01 to NPK. The funders had no role in study design, data collection and analysis, decision to publish, or preparation of the manuscript.

### Grant Disclosures

The following grant information was disclosed by the authors:
NIH: Al065728-01, AI55397.

### Competing Interests

Nancy Keller is an Academic Editor on the Editorial Board of PeerJ.

### Author Contributions

- Jonathan M. Palmer conceived and designed the experiments, performed the experiments, analyzed the data, wrote the paper.
- Jin Woo Bok conceived and designed the experiments, performed the experiments, analyzed the data.
- Seul Lee performed the experiments.
- Taylor R.T. Dagenais performed the experiments, analyzed the data.
- David R. Andes and Dimitrios P. Kontoyiannis contributed reagents/materials/analysis tools.
- Nancy P. Keller conceived and designed the experiments, analyzed the data, contributed reagents/materials/analysis tools, wrote the paper.

### Animal Ethics

The following information was supplied relating to ethical approvals (i.e. approving body and any reference numbers):

William S. Middleton VA and University of Wisconsin Animal Care Committees according the animal use protocol MV2344.

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
