# Peer review of "Loss of CclA, required for histone 3 lysine 4 methylation, decreases growth but increases secondary metabolite production in Aspergillus fumigatus"

_PeerJ, doi:10.7717/peerj.4_

## Round 0.1 · original submission · Minor Revisions

· Academic Editor

Minor Revisions

Reviewer # 1 has some concerns about the novelty of this work and suggests further experiments that could be undertaken to enhance and extend the study. As novelty is not a requirement for PeerJ, and both reviewers find the work to be done to a high standard, I leave it with you to decide whether to incorporate further work as suggested.

Both reviewers have questions about some of the experimental design and/or interpretation. Please ensure you address these in your revised manuscript.

Reviewer 1 ·

Basic reporting

The manuscript describes the knock-out of two Aspergillus fumigatus genes, one coding for a member of the histone 3 lysine 4 methylating COMPASS complex (CclA) and the other for a protein similar to H3K4 demethylase (HdmA). A similar knock-out of cclA in Aspergillus nidulans has been previously reported, whereas no studies have been performed with the hdmA gene. The null mutant strains were examined in respect to growth rate, resistance of 6-azauracil, histone methylation, production of secondary metabolites (SMs) and virulence in animal and insect models. The hdmAΔ mutant proved indistinguishable from an isogenic wild-type control in all tests performed. The cclAΔ mutant showed reduced growth rate, deficiency in tri- and di-methylation of H3K4 and relatively increased production of SMs. These results practically duplicate those already found in A. nidulans. None of the two mutant strains showed altered A. fumigatus pathogenicity in animal and insect models, as might have been hoped.

Major Point
The manuscript is extremely well written and easy to read. The experiments are nicely performed and the conclusions drawn reflect the results obtained. However my main concern is that this is not a complete story and should rather be the beginning of a more in-depth study. For example, the authors state that there is putative protein encoded by Afu5g03430 which might in fact be the major demethylase of H3K4 in the aspergilli, but surprisingly they have not experimentally followed their observation. Another example showing that this is not full story is the very preliminary analysis of the identity of the SMs produced in the cclAΔ mutant. Given that PeerJ is a new journal it also depends on the editors whether they wish to publish complete and original stories or well performed experiments that do not lead to novel knowledge. Maybe a specialized fungal journal for short reports will be more proper for publication of the present work as it is.

Some minor points
1. The 6-azauracil test as presented and the reasoning on its action are not convincing to me. I would rather prefer to see real growth tests under various conditions. Is the wt A. fumigatus sensitive to 6-azauracil? At which concentration has this drug an effect? A mutant that affects significantly growth, such as cclAΔ, can be affected indirectly into several metabolic pathways and transport activities. The authors could have easily tested whether specific uracil transporters (homologues of FurD of A. nidulans) have been affected in the cclAΔ mutant, in order to strengthen their reasoning that 6-azauracil sensitivity reflects changes in chromatin.
2. Figure 2 could be reduced. Most articles describing knock-out do not show expended southern blot analysis, at least as primary figures.

Experimental design

1. The 6-azauracil test as presented and the reasoning on its action are not convincing to me. I would rather prefer to see real growth tests under various conditions. Is the wt A. fumigatus sensitive to 6-azauracil? At which concentration has this drug an effect? A mutant that affects significantly growth, such as cclAΔ, can be affected indirectly into several metabolic pathways and transport activities. The authors could have easily tested whether specific uracil transporters (homologues of FurD of A. nidulans) have been affected in the cclAΔ mutant, in order to strengthen their reasoning that 6-azauracil sensitivity reflects changes in chromatin.
2. Figure 2 could be reduced. Most articles describing knock-out do not show expended southern blot analysis, at least as primary figures.

Validity of the findings

see basic report

Additional comments

no comment

Reviewer 2 ·

Basic reporting

no comments

Experimental design

no comments

Validity of the findings

No comments

Additional comments

This manuscript described an investigation into the roles of CclA and HdmA in Aspergillus fumigatus. Deletion of HdmA does not significantly alter any phenotypes that the authors examined, but the deletion of CclA renders the strain slow-growing. The strain also shows enhanced SM production in vitro. Interestingly, the cclA deletion mutant displays the same level of virulence in the toll fly model and a mouse model.
A few specific comments:
1. It is not clear to me based on what is presented in the result that the cclA mutant is defective in transcription. The authors implied in the result section (page 10) that the cclA mutant is specifically sensitive to 6AU. The authors need to list all the compounds tested and the mode of action of the tested compounds. That way, the specificity to transcription can be assessed.

2. It is interesting that the growth temperature significantly affects the amount of SM produced by the cclA mutant. Have the authors checked the histone 3 lysine 4 methylation patterns under these conditions in the cclA mutant and see if there is any correlation?

---

## Round 0.2 · accepted · Accept

· Academic Editor

Accept

Thank you for taking on board the comments by the reviewers and modifying the manuscript.